# On the Analysis and Reproduction of "Post-hoc Concept Bottleneck Models" with an Extension to the Audio Domain

## Abstract

Although deep neural networks are powerful tools, they are yet considered "black boxes". With the proliferation of AI models, the need for their interpretability has increased. One way to improve the interpretability of deep neural networks is to understand their decisions in terms of human-understandable concepts. *Concept Bottleneck Models* (CBMs) aim to achieve this goal by using embedding representations of concepts into the model, providing explainability into the decisions that a network makes. However, CBMs have various limitations concerning training efficiency and task applicability. The authors of the paper *Post-hoc Concept Bottleneck Models* (PCBMs) provide a novel approach to creating CBMs in a more efficient and generalizable way. In this paper, we evaluate their claims, namely, that PCBMs can be trained using any pre-trained neural network and that PCBMs offer interpretability without sacrificing significant performance. To do so, we not only attempted to **reproduce** the original paper results but also **extended** the approach into the audio domain. Our results show good alignment with the original paper but further analysis revealed some problems PCBMs may have, namely, challenges in getting a suitable list of relevant human-understandable concepts for a given task, and potential misalignment between concept encoders and input feature encoders. The code for our paper can be found at `https://anonymous.4open.science/r/-354E/`

## 1 Introduction

Deep neural networks have become widely applied to real-life problems, making decisions in areas that have high stakes such as health, finance, and policing (Adadi & Berrada, 2018). This calls for improving the interpretability of the models in use, to confirm the decisions made by the models are indeed reliable and transparent. To make deep learning more interpretable, analysis methods based on human-understandable concepts were introduced, helping us see how such concepts are generated and used in neural networks – this includes *Concept Bottleneck Models* (CBMs) (Koh et al., 2020). CBMs are a neural network-based method, trained end-to-end to learn to predict a set of human-understandable concepts in the input and then linearly combine the vector representations of concepts to predict the labels in a classification task. By doing so, we can explain the decision made by the model by looking at the concepts which have the highest importance, and likewise understand the mistakes it makes. The paper under investigation here introduced *Post-hoc Concept Bottleneck Models* (PCBMs) (Yuksekgonul et al., 2022), where instead of training a model to recognize concepts and perform classification tasks in one go, PCBMs build simple concept-based models using a pre-trained neural network for classification. This was intended to help overcome the following limitations of CBMs:

1. **Need for labeled data**: CBMs require access to data-concept pairs, which may require manual labeling. Using PCBMs, one can use multi-modal models to map data to concepts directly.

2. **Performance**: The accuracy CBMs generally achieve is lower than that of models trained without being constrained by concepts. With PCBMs, one can combine information from the pre-trained original model with the concept-based model via a separately trained residual connection, thereby improving the prediction accuracy.

3. **Model editing**: Since CBMs are trained end-to-end, it is unspecified how global editing may be achieved in CBMs. In PCBMs however, manual and automated pruning of concepts was introduced and was shown to improve model performance by Yuksekgonul et al. (2022).

In this paper, we test the following claims from the original paper by Yuksekgonul et al. (2022):

- *Claim 1*: PCBMs can be applied to a wide range of neural networks [1],

- *Claim 2*: PCBMs achieve high prediction accuracy comparable to the models they are based on,

- *Claim 3*: PCBMs offer interpretability benefits.

To test these claims, we first **reproduce** the results of the experiments done in the original paper in the domain of image classification. We examine whether we can indeed turn the 'backbone' pre-trained models introduced in the paper into PCBMs, whether they achieve comparable performance to original models, and whether the most important concepts used by PCBMs are in line with human domain knowledge or common sense. Since these previous experiments were performed exclusively in the image classification domain, we conduct experiments to test whether these claims are **generalizable** in another domain, namely audio classification.

## 2 Methods

This section outlines our approach in replicating the original study as well as our extension to the audio domain. The tasks we focus on here are image classification (reproduction of the original results) and audio classification (testing the generalization of the stated claims).

### 2.1 Backbone models

We start by choosing the pre-trained models ('backbone models') we would like to interpret using the PCBM method. For image classification, we use the ones specified in the original paper, which were CLIP-ResNet50 (Radford et al., 2021), ResNet18 (He et al., 2016), and the Inception model (Szegedy et al., 2014). These CNN-based models were chosen to test PCBMs in different challenging image classification tasks. For audio classification, we used one CNN-based architecture, ResNet34 (He et al., 2016), and a transformer architecture, HTS-AT (Chen et al., 2022).

In both domains, each backbone was tested with one or more datasets, detailed in Section 2.4. For both models, we first pre-process the audio into log-Mel spectrograms such that it can be fed into ResNet34's image encoder as well as HTS-AT's transformer-based audio encoder.

### 2.2 Concept libraries

To build the PCBMs, we first need to create a 'concept library' for a given dataset – this is just a set of concepts that might be relevant to interpreting the decisions of the model. For the image classification tasks, we use the knowledge graph ConceptNet (Speer et al., 2017) to obtain concepts related to the targets.

For audio classification, we constructed concept libraries on our own by asking GPT-4 via ChatGPT (OpenAI, 2023) to generate five concepts for each class. Our prompt starts with "Is there an audio conceptNet?". Then we repeated the following request for 50 times (one for each class) " Pulling concepts for *class*". Finally, we added, "can you give me 50 concepts in a structured way to me". We decided on this due to the fact that concepts output by ConceptNet generally describe visual cues, which may not match well with audio input.

---

[1]This is the modified version of the original claim: '*One can turn any neural network into PCBMs*', which is quite a strong claim from the authors of the original paper. Here we are not trying to prove the claim but evaluate whether we can turn different kinds of neural networks into PCBMs, e.g. those from audio domains

### 2.2.1 Concept Activation Vectors

*Concept Activation Vectors* (CAVs) (Kim et al., 2018) are vectors that somewhat represent a particular concept, chosen from the concept library. A set of inputs are chosen that are labeled to contain this concept, and likewise, a set is chosen that does not contain said concept. We can then learn a linear decision boundary between regions in the embedding space of an encoding network that have and do not have the concept, using the data points in the positive and negative sets as inputs. The CAV is acquired by finding the vector normal to the linear classification boundary trained, which, in all cases mentioned below, is learned with a linear Support Vector Machine (SVM). In the case where we do not have concept annotations, we extracted concepts via the true class label from a knowledge graph (i.e. ConceptNet), and used multi-modal models such as CLIP to automatically extract the corresponding embeddings as concept vectors. We explore both these methods in our reproductions. These concept vectors are then concatenated vertically, with each row being one concept vector, to form a concept 'subspace'.

To extend the method to work with audio, where we did not use manually labeled concepts, we made use of the contrastive language *audio* pertaining (CLAP) model (Wu et al., 2023). CLAP was inspired by CLIP and works similarly by consisting of both an audio and text encoder model that, capable of producing aligned embeddings for audio and text data. For our experiments, the audio encoder utilizes the HTS-AT (Chen et al., 2022) transformer architecture within the pre-trained CLAP model. With CLAP, different concepts for different classes are given as input into the text feature extractor.

### 2.3 Training PCBMs and PCBM-hs

With the concept subspace prepared, we then extract embeddings from the penultimate layer of the backbones and project them onto the concept subspace. These projections are subsequently used to train sparse linear classifiers with concepts as features. With $x$ being the samples of the dataset $\mathcal{D}$, PCBMs are linear classifiers $g\left(f_{\boldsymbol{C}}(x)\right) = \boldsymbol{w}^T f_{\boldsymbol{C}}(x) + b$, where $f_C = \text{proj}_C f(x)$ are the projections of data embeddings $f(x)$ (where $f$ is some embedding network) onto the concept subspace $C$, and $y$ are the targets. To train PCBMs, we use the following optimization objective:

$$\min_{g} \mathbb{E}_{(x,y)\sim\mathcal{D}}\left[\mathcal{L}\left(g\left(f_{\boldsymbol{C}}(x)\right), y\right)\right] + \frac{\lambda}{N_c K}\Omega(g)$$

Here we minimize the loss function $\mathcal{L}$ (e.g. cross-entropy loss) with a regularization term where $\Omega(g) = \alpha\|\boldsymbol{w}\|_1 + (1-\alpha)\|\boldsymbol{w}\|_2^2$ is a complexity measure to regularize the model for sparsity, namely, the elastic-net penalty parameterized by $\alpha$. $N_c K$ is a normalization term (the number of concepts and classes respectively) and $\lambda$ is the regularization strength.

Following the original paper, we also build *Hybrid PCBMs* (PCBM-h) to fit the residuals by training the sum of the linear classifiers described above and the embeddings of the backbone models. This can be written out as follows:

$$\min_{r} \mathbb{E}_{(x,y)\sim\mathcal{D}}\left[\mathcal{L}\left(g\left(f_{\boldsymbol{C}}(x)\right) + r(f(x)), y\right)\right]$$

Here $r : \mathbb{R}^d \to \mathcal{Y}$ is the residual predictor, which is implemented as a linear model $r(f(x)) = w_r^T f(x) + b_r$, $d$ is the size of the embeddings and $\mathcal{Y}$ is set of targets in the task-specific encoding (e.g. one-hot). While training the residual predictor, the interpretable predictor $g$ is trained independently and kept fixed. To observe model performance in the absence of the residual predictor, we can drop $r$.

### 2.4 Datasets & Evaluation

To evaluate whether PCBMs and PCBM-hs offer comparable performance to the original models, we replicated a selection of experiments from the original paper. Specifically, we used the datasets CIFAR10, CIFAR100, CUB, and HAM10000. Due to limitations in code availability, specific implementation details,

we decided to discard SIIM-ISIC, which were part of the original study. We then used the datasets ESC-50 (Piczak, 2015) and FSDKaggle2018 (Fonseca et al., 2018) to test audio PCBMs. For each dataset, we conducted 10 tests using different seeds.

To check the interpretability of the PCBMs, we use the same analysis method as in the origin paper as well – we extract concept weights from the sparse linear classification model and identify influential concepts for each class by ranking these concepts based on their magnitude and selecting the top-$N$ concepts that exhibited the greatest influence on the classification process. We then verified whether these were relevant to the target class (see Section 3 for details). Below is a summary of the setup for each dataset that was used in the evaluation:

**CIFAR10, CIFAR100 (Krizhevsky et al., 2009)**: For these two datasets, the backbone model was a pre-trained CLIP-ResNet50 (He et al., 2015). It classifies images into unseen categories using pre-trained knowledge (Wang et al., 2019). Both datasets incorporate 170 concepts following Abid et al. (2022), specifically employing BRODEN concepts. The original training and test splits of both CIFAR datasets were used along with the same training parameters as the original authors.

**CUB (Wah et al., 2011)**: In the CUB dataset, as was also detailed by He & Peng (2020), we address a 200-way bird identification task employing a ResNet18 model specifically pre-trained for the CUB dataset. The preprocessing of input data adheres to the methodologies outlined in the original paper. We adopted the same training and validation splits, utilizing the 112 concepts described in Koh et al. (2020).

**HAM10000 (Tschandl et al., 2018)**: For the HAM10000 dataset, we utilized an Inception model specifically trained to analyze dermoscopic images from the HAM10000 dataset. This model aims to determine whether a skin lesion is benign or malignant. It incorporates the eight concepts outlined in the Derm7pt criteria and adheres to the experimental settings presented in the study by Lucieri et al. (2020).

**FSDKaggle2018 (Fonseca et al., 2018)**: We fine-tuned a ResNet34 model ourselves for the audio domain on the training subset of this data set. The ResNet34 was pre-trained on the ImageNet1k dataset (Deng et al., 2009). Since this model requires images as input, we pre-processed the audio wave data to log-Mel spectrograms. We use 232 audio concepts generated by CLAP.

**ESC-50 (Piczak, 2015)**: For the ESC-50 environmental audio classification data set, we also used the ResNet34 pre-trained on the FSDKaggle2018 data set described previously. Additionally, we evaluate PCBMs using CLAP's HTS-AT transformer backbone with pre-trained weights on this data set (Chen et al., 2022). The concepts were fed into the CLAP text encoder to create the concept embeddings.

### 2.5 A Note on Adapted Methodology for CLIP

While the original CLIP model relied on zero-shot classification, directly associating images with textual class descriptions using pre-trained embeddings, our adjusted approach involves training specialized classifiers atop CLIP's image features for tasks such as binary and multi-class classification. This adjustment serves to augment the performance of the baseline CLIP model and provides more flexibility in handling specific classification tasks.

## 3 Results

This section lays out the results attained from both the reproducibility studies conducted (Section 3.1) and our own extension into the audio domain (Section 3.3), as well as relevant analysis. When applicable, results in the tables indicate the mean result over 10 seeds, $\pm$ the standard deviation. Additionally, there are a number of ways in literature to decide if results have been successfully reproduced (e.g. looking at trend lines instead of absolute values, as in Garcarz et al. 2023, subjectively deciding closeness, as in Don et al. 2023, or only looking closely at the verification of claims and not values, as in Baratov et al. 2023). In this paper, we consider an accuracy value to be reproduced successfully if it lies within 5 percentage points of the original (as was done previously in Sun et al. 2023). Though a fuller statistical analysis in future works may be needed for robustness, we believe this is not required for the purposes of verifying the claims and stated values of the original paper.

### 3.1 Image classification: Performance

Our reproduced results closely match those of the original paper (Table 1), which makes it reasonable to assume that their method is reproducible. The only results we have not been able to reproduce so far are those of the original models, where we achieve lower accuracy than stated in the original work.

| | Model | CIFAR10 | CIFAR100 | CUB | HAM10000 | COCO-Stuff |
|---|---|---|---|---|---|---|
| | Original Model | 0.888 | 0.701 | 0.612 | 0.963 | 0.770 |
| Original | PCBM | $0.777 \pm 0.003$ | $0.520 \pm 0.005$ | $0.588 \pm 0.008$ | $0.947 \pm 0.001$ | $0.741 \pm 0.002$ |
| | PCBM-h | $0.871 \pm 0.001$ | $0.680 \pm 0.001$ | $0.610 \pm 0.010$ | $0.962 \pm 0.002$ | $0.768 \pm 0.01$ |
| | Original Model | 0.879 | 0.691 | 0.611 | 0.891 | 0.793 |
| Ours | PCBM | $0.814 \pm 0.001$ | $0.536 \pm 0.001$ | $0.579 \pm 0.005$ | $0.949 \pm 0.001$ | $0.713 \pm 0.006$ |
| | PCBM-h | $0.885 \pm 0.006$ | $0.688 \pm 0.003$ | $0.580 \pm 0.003$ | $0.961 \pm 0.003$ | $0.757 \pm 0.011$ |

Table 1: **Reproduction of PCBM performance results**. "Ours" refers to our attempts at reproducing the results of the "Original" paper (Yuksekgonul et al., 2022). The performance of our baselines, PCBM and PCBM-h models is in line with their results.

| | Model | CIFAR10 | CIFAR100 | COCO-Stuff |
|---|---|---|---|---|
| | Original Model (CLIP) | 0.888 | 0.701 | 0.770 |
| Yuksekgonul et al. (2022) | PCBM & labeled concepts | $0.777 \pm 0.003$ | $0.520 \pm 0.005$ | $0.741 \pm 0.002$ |
| | PCBM & CLIP concepts | $0.833 \pm 0.003$ | $0.600 \pm 0.003$ | $0.755 \pm 0.001$ |
| | PCBM-h & CLIP concepts | $0.874 \pm 0.001$ | $0.691 \pm 0.006$ | $0.769 \pm 0.001$ |
| | Original Model (CLIP) | 0.879 | 0.691 | 0.793 |
| Ours (Reproduction) | PCBM & labeled concepts | $0.814 \pm 0.001$ | $0.536 \pm 0.001$ | $0.713 \pm 0.006$ |
| | PCBM & CLIP concepts | $0.841 \pm 0.002$ | $0.572 \pm 0.004$ | $0.725 \pm 0.003$ |
| | PCBM-h & CLIP concepts | $0.865 \pm 0.002$ | $0.675 \pm 0.002$ | $0.760 \pm 0.001$ |

Table 2: **Reproduced results with CLIP concepts.** Here concept descriptions generated by CLIP are used to generate the concept subspace, which alleviates the reliance on concept annotations. Our PCBM and PCBM-h models align with the original paper's results.

| Model | Mean Difference |
|---|---|
| Original Model | $0.0138 \pm 0.0314$ |
| PCBM | $0.0036 \pm 0.0220$ |
| PCBM-h | $0.0040 \pm 0.0155$ |
| PCBM & CLIP | $0.044 \pm 0.017$ |
| PCBM-h & CLIP | $0.011 \pm 0.003$ |

Table 3: Performance Comparison

Our reproduced results with CLIP concepts, i.e. using unlabeled data with multi-modal models for generating concept vectors, (Table 2) also closely match those of the original paper, which makes it reasonable to assume that they are reproducible as well.

### 3.2 Image classification: Interpretability

Figure 1 from the original paper displays the ranked magnitude of concept weights in PCBMs on classification tasks, highlighting the model's interpretability. Our reproduction of this is in Figure 2. After ranking concepts by the magnitude of their respective weights and identifying the top-$N$ concepts that significantly influence classification ($N = 3$ in the original, 5 in our reproduction), we demonstrate comparable findings and interpretability between our study and the original. The observed variation in the range of concept weights between our study and the original paper can be attributed to differences in the regularization

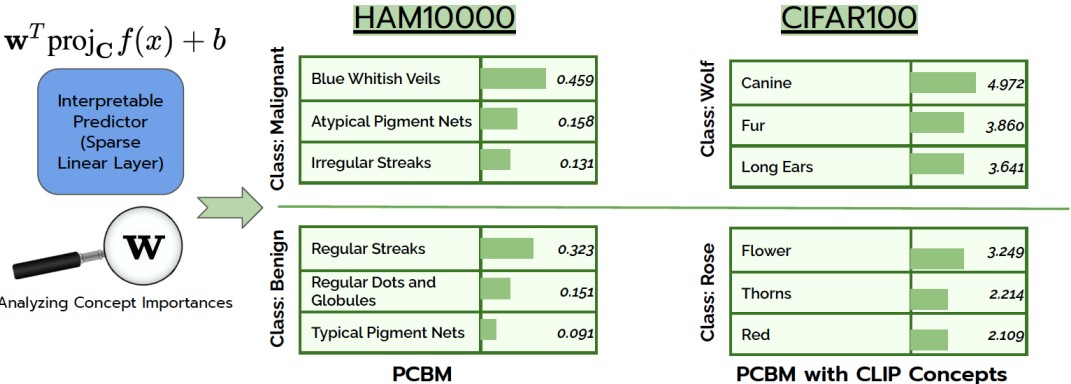

Figure 1: **Concept weight interpretation (original).** The concepts with the highest weights in the PCBM's linear classification layer align well with the class labels. This figure is reproduced from Yuksekgonul et al. (2022).

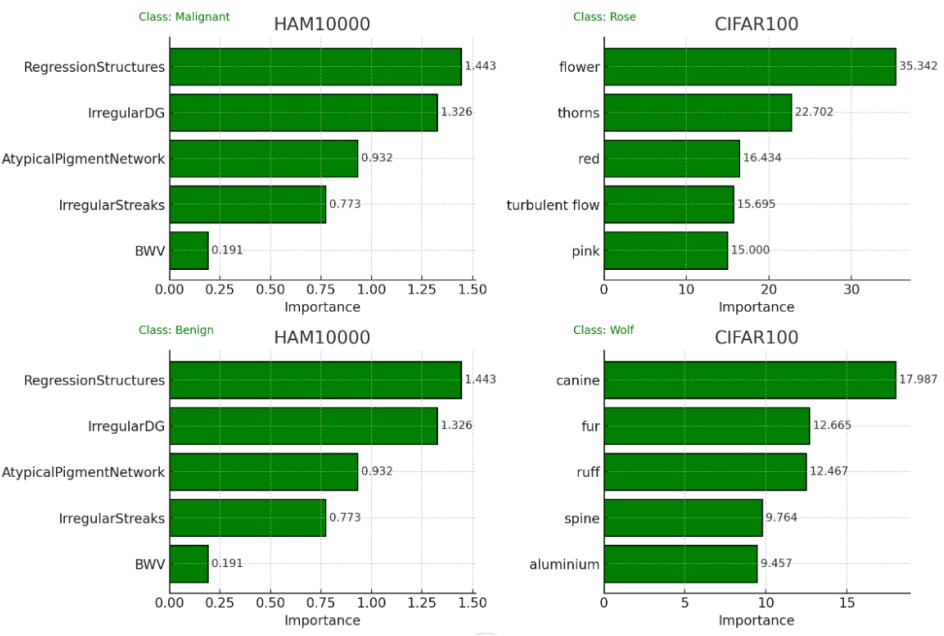

Figure 2: **Concept weight interpretation (reproduction).** In our reproduction, we also observe that the concepts match the class labels well.

parameter settings for both PCBMs and SVMs. This parameter significantly influences the magnitude of weights, though often not the ranking of concepts themselves.

Additionally, in reproducing the results for the HAM1000 task, our PCBM produces the same weights for both classes, unlike in the original paper. In our attempts to reproduce these results, we treated the skin lesion classification task as a binary classification task (either benign or malignant), which results in the PCBM being a single logistic regression model, with a single weight matrix for both classes. This is also the case in the provided implementation by the authors. Thus it is unclear to us how the original authors were able to produce two sets of differing weights.

## 3.3 Audio classification: Performance

Using the experimental setup described in Section 2, we get promising results with PCBMs, as can be seen in table 4. For both backbone models, the audio PCBM only suffers a small loss in performance compared to our baseline on the ESC-50 environmental audio classification task. We also see that nearly half of the deficit is recovered by using the Hybrid-PCMB model. For the FSD2018Kaggle data set, we observe a similar loss in performance with our PCBM model with the ResNet34 backbone. Here, again, almost all of the deficit can be recovered with the PCBM-h model. Thus, we see that claims 1 and 2 are both supported when generalizing this method to a new domain, namely, that all networks can become PCMBs and that PCBMs achieve high accuracy comparable to the original model.

| Backbone model | Model | ESC-50 | FSD2018Kaggle |
|---|---|---|---|
| ResNet34 | Original model | 0.733 | 0.855 |
| | PCBM | $0.701 \pm 0.012$ | $0.825 \pm 0.009$ |
| | PCBM-h | $0.713 \pm 0.000$ | $0.846 \pm 0.001$ |
| HTS-AT | Original model | 0.940 | N/A |
| | PCBM | $0.935 \pm 0.010$ | N/A |
| | PCBM-h | $0.940 \pm 0.000$ | N/A |

Table 4: The PCBM trained in the audio domain performs almost as well as the baseline ResNet34 while the PCBM-h decreases the gap in accuracy further. For the transformer HTS-AT backbone, the PCBM closely approaches baseline performance, with PCBM-h matching the original model.

## 3.4 Audio classification: Interpretability

As with the image interpretations, after training our audio PCBM based on the ResNet34, we ranked concepts based on their weights, and observed the highest weights in the linear classification layer to the most influential $N = 5$ concepts for a given class label. As can be seen from Figure 3, the concepts do not intuitively match the label very well.

The HTS-AT PCBM results show a significant increase in performance on the ESC-50 dataset, illustrated earlier in Table 4. In Figure 3, we interpret the weights of this PCBM. We can see that concepts used for classification match the predicted class much more closely compared to the ResNet34 PCBM concepts. We believe that the discrepancy in the interpretability lies primarily in the different ways embeddings are created in both models.

## 4 Discussion

In this section, we discuss various aspects of our reproduction and analysis, and justify whether or not the claims of Yuksekgonul et al. (2022) are affirmed by our study.

## 4.1 Reproducibility of the Original Results and Generalization

In Sections 3.1 and 3.2, we reproduced the majority of results from the original paper, suggesting the reproducibility and efficacy of PCBMs as a method. We saw that there was no significant deviation to the

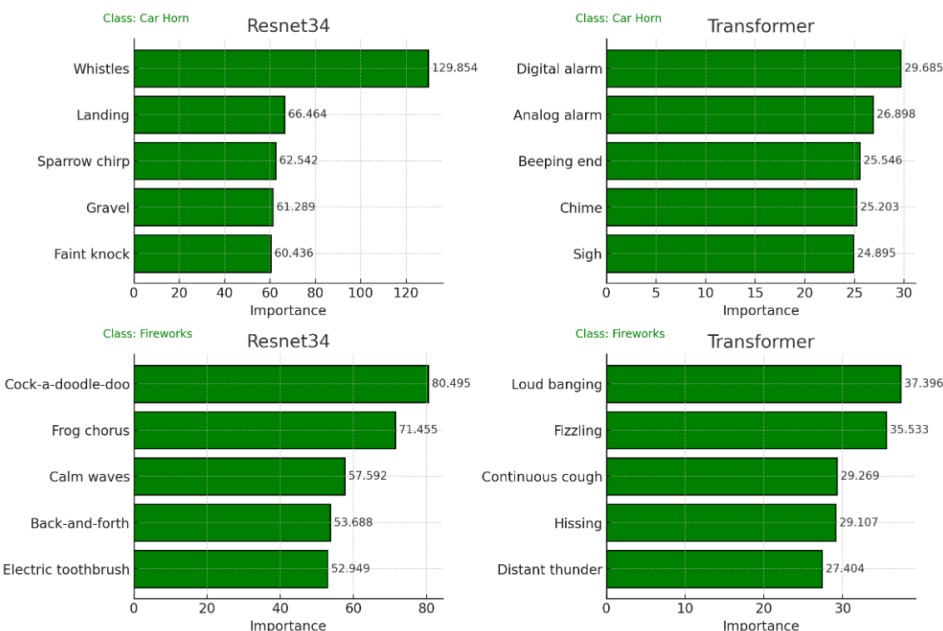

Figure 3: **Concept Weights Audio PCBM.** Even though we are using the same concept bank generated by GPT4, the concepts with the highest weight produced by the PCBM with the HTS-AT backbone make way more sense from a human point of view. The corresponding concepts with the highest weight from the ResNet PCBM do not seem to match well at all with the class label.

results. As described in Section 3, we defined a successful reproduction of a value as when it is within 5 percentage points of the original.

Additionally, by using a similar pipeline to the original for the audio domain, we can create PCBMs for audio classification and achieve high-accuracy results with somewhat interpretable results.

Thus, overall, we can say that claims 1 and 2 are supported, that a wide range of neural networks can be turned into a PCBM with no significant change in performance. Claim 3 about the interpretability of PCBMs is only partially justified, as we have shown in the audio domain. We discuss this point further in Section 4.4.

## 4.2 Deviation of Results from Original Paper

In general, replicating the performances of each PCBM and PCBM-h for the majority of datasets was facilitated by the comprehensive code provided by the original paper. Except for the ISIC datasets, we found the replication easy, affirming the reproducibility of the original study's methodology for most tested scenarios.

Though we claim to have reproduced the original paper's results to a relatively small margin (within 5 percentage points for each result), our results do deviate from theirs slightly. This discrepancy in model accuracy could stem from several factors. First, the original study may have utilized a specific version of the CLIP model or different pre-processing steps that were not explicitly documented. Subtle variations in image scaling, normalization, or data augmentation can have a profound impact on model performance. Another potential reason could be the difference in utilized pre-trained model weights. If the original study used pre-trained models with certain checkpoints, the absence of these specific weights in our replication would lead to discrepancies. Lastly, variations in hyperparameter settings such as batch size, learning rate, or even random seed during training can introduce non-negligible differences in outcomes. A more thorough disclosure of the experimental setup in the original paper would greatly aid in addressing these issues. Our hyperparameters are disclosed in Appendix A.

In future work, a detailed study controlling these factors could pinpoint the source of the performance gap. Moreover, direct communication with the original authors could clarify unreported details crucial for an accurate replication. Though we attempted this (and a follow up), no response from the original authors was received by the time of writing this paper.

### 4.3 Impact of Concept Banks

While creating audio PCBMs, we found that a suitable concept bank can be difficult to construct for domains like audio classification. This is because different senses can encode very different information therefore leading to very different concepts (Ezgi Mamus & Majid, 2023). For example, we can see a cat has whiskers, four legs, sharp claws, etc., but we cannot *hear* those concepts from a cat. Instead, we may hear auditory features such as high-pitched, soft, intimate, etc.

We also need to consider how the multi-modal models are utilized. Specifically, we need to pay attention to what texts are used in image-text pairs and avoid choosing a concept bank with vocabulary that is not covered by the text-encoder in the training of these multi-modal models.

With our list of concepts, our audio PCBMs achieved performance very close to the baseline model. When we look at the weights of the most impactful concepts, however, we find that the concept descriptions do not match the classes well. In the future, we wish to extract audio concepts in a structured way similar to the original paper using the AudioSet (Gemmeke et al., 2017) ontology for sounds. We believe this would yield a more meaningful set of concepts and help the model learn generalizable representations. Using a larger training data set could also help our model better learn appropriate concepts, such as experiments with additional audio data, thereby also including domains outside of environmental audio and perhaps even more specialized ones (e.g. medical audio data).

### 4.4 Alignment between Embeddings and Concept Subspace

In the original paper, the authors proposed two ways to generate concept subspaces. One is through CAVs, and another is by using multi-modal models. In CAVs, the embeddings generated by backbone models are used to create the concept subspace. While using multi-modal models, the shared embeddings of the image encoder and text encoders are used for creating the concept subspace. For example, when creating audio PCBMs, we used ResNet as a backbone model to create embeddings, while the audio encoder in CLAP is transformer-based. Different initializations and architectures may lead to different encodings, and depending on the difference in the size of the embedding spaces, we might lose a significant amount of information when projecting the embeddings to the concept subspace.

This means the connection between text and audio representations in CLAP with a transformer audio encoder may not apply to the audio representations from ResNet. This can in the end cause a mismatch between the audio inputs and concepts, leading to incoherent or irrelevant concepts being used to explain the outputs of a model. We believe having the baseline model and audio encoder in CLAP aligned or a similar size would provide us with a meaningful and interpretable set of concepts for a given input.

Later on, we changed the backbone model to a transformer similar to the encoder in CLAP. From our results, this change increased both accuracy and interpretability.

From this, we would like to re-evaluate the claim the authors made that one can turn any neural network into PCBMs without labeled data. Using multi-modal models to construct concept subspace indeed makes it much easier to build a PCBM, so we do not have to manually annotate concepts. However, it is not yet straightforward to adopt a multi-modal model because one needs to make sure the multi-modal model uses the same encoder model as the backbone, meaning we might need to fine-tune it to retain most of its original performance.

### 4.5 Computational Resources

Since the main premise of the original paper is about adapting pre-trained models only by adding a single trainable linear layer, it is quite light on resource needs. As we did not find suitable backbone models pre-

trained specifically on spectrogram data, we decided to fine-tune a pre-trained model. In total, including computing the concept vectors, fine-tuning and testing of the ESC-50 and FSDKaggle2018 datasets, our models ran for a total of less than 60 minutes on a Nvidia RTX 4090 GPU, which is estimated to be 0.15kg $CO_2eq$ in total. Training of the PCBMs was conducted on CPUs for all data sets. This indeed shows an advantage of PCBMs in terms of computational resources.

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

## A  Hyperparameter Configuration

All settings and hyperparameters not mentioned here (and are generally not relevant to the reproduction of PCBMs) may be found in our code, mentioned in the abstract of this paper.

In our general PCBM experimental setup, the elastic net sparsity ratio, denoted as $\alpha$, was consistently set to 0.99 across all models. We employed a Stochastic Gradient Descent Classifier (SGDClassifier) for training. This classifier was configured with a maximum of 10,000 iterations and was augmented by an elastic net penalty. We adopted the Adam optimization algorithm for components of the hybrid model utilizing PyTorch. The learning rate (`-lr`) for this algorithm was set to $1 \times 10^{-3}$. The training epochs are set to be 10, with a batch size of 64 and a worker count of 4. For the PCBM-h experiment, the training

epochs are set to 20, with a batch size of 64 and a worker count of 4. The learning rate (`-lr`) are set to $1 \times 10^{-2}$. The l2-penalty is set to $1 \times 10^{-3}$.

The parameters used for the SVM to obtain the concept vectors are the same for all concept banks. We use a regularization parameter of 0.1. For the number of samples for each of the sets that contain or do not contain the concepts, we use different values though, for CIFAR10 and CIFAR100 and HAM10000 we use 50 samples, whereas for CUB we use 100.

For these two multi-modal models, for CIFAR10 and CIFAR100, the model backbone was a CLIP-ResNet50. The regularization strength, represented by `-lam`, was calibrated through a grid search to a value of $1 \times 10^{-5}$. with 50 samples.

