# OpenReview forum: "On the Analysis and Reproduction of "Post-hoc Concept Bottleneck Models" with an Extension to the Audio Domain"
_TMLR — Rejected by TMLR_

### Review · Reviewer_KhdV · 2024-03-26

**Summary Of Contributions:**

The main goal of the paper is to reproduce previously published paper "Post-hoc concept bottleneck models" with sharing best practices, the claims that can be reproduced following the paper and published code for that paper, the claims which cannot be reproduced. Moreover, authors apply the paper idea and methodology to audio domain (as before it is shown only for vision data) and extend results and claims to audio datasets. In summary, authors are able to reproduce part of the results, they discuss what and why they were not able to reproduce (but the main algorithm was reproduced), they apply technique to audio domain and show that it also works there, however there is an issue of getting the proper interpretability which can be related to the issue of the backbone and pre-trained model discrepancy in the feature extraction -- authors discuss this in the last section as the issue of the multi-modal representation and proper stacking the models in this case.

**Audience:**

Yes

**Broader Impact Concerns:**

The work is important on getting more controlled models via interpretability, showing that prior methods can be extended to other domains (from vision to audio), and getting reproducible research results and claims. I do not have any concerns with the current form of the paper.

**Claims And Evidence:**

Yes

**Requested Changes:**

Please find below suggested improvement for the text and some suggestions to strength the paper's results:
- clarity
  - page 2 Claim 1 formulation: "any neural network" - this is too strong claim, as we cannot brute force all possible models, I would smooth formulation. Otherwise did original paper had math proof (which I guess also impossible to have with the current math we have)
  - page 2 claim 2 formulation: "original model" - it is not clear for the entire paper what is exactly original model - is it model we train which later we try to change to the PCBM by changes the arch and loss and training from scratch? Maybe it is CBM model?
  - sec 2.1 "in both domains" - you are speaking about audio, why here both domains? fix paragraphs on speaking only about vision, then about audio, then about both together.
  - through the paper for me it is not clear which parts are reproduction, which parts some modifications. Better to be strict everywhere on this and have bold text e.g. About audio part - be also clear in text that all things are new and not reproduction. Right now it is mixed a bit in the text.
  - sec 2.3 f(x) definition is not given
  - sec 2.4 better to discuss why you decided to discard COCO right away in the section. Also is the size of COCO benchmark is larger? I wonder if there are issues on reproduction the scaled version, when data/model size is increased.
  - what is exactly "original model"? from text it is hard to get
  sec 3.2 last paragraph - highlight it more - it is important thing you checked!
  - Table 3: first sentence in the caption is really weird, I don't see the "half the accuracy loss"
  - page 8 top paragraph - "any neural net can be" - this is too strong claim, we never can show it (except from theoretical proof). So you need to smooth formulation.
  - Sec 4.3 last paragraph "for sounds We believe" - missed dot (I didn't see any other typos :) ).
  - Sec 4.5 - "spectrogram image data"  - remove image, why it is there?
  -  Appendix B "It is repeated 50 times" - I didn't get what is exactly repeated. Why also in the prompt it is underscore in the names of classes? Did you try to remove it to get better output as we know things are sensitive to the punctuation formatting in the prompts.
- methodological
  - why did you use Imagenet as pretrained model to finetune on audio dataset? I think this is really inappropriate comparison.
  - why for the second dataset of audio (ESC 50) you use pretrained model to be the one finetuned on FSDKaggle2018?
  - the std for the same codebase is small, but reproduction between codebases is not within this std, so discussion on why there are still discrepancies would be beneficial
  - figure 3 - is it because we used pretrained model on Imagenet?
  - why not using AudioSet for experiments? It is more popular dataset, maybe the issue it is big for resources you have to be able to train on it? (then it is ok if not enough resources).
  - why any model trained on audio / speech is not used as pretrained model? e.g. trying wav2vec 2.0 or some other things from speech domain - yep it is OOD for the audio classification but closer in terms of representation that imagenet models.
  - some prior works on audio datasets should be provided to get sense how the original model performance or proposed model perform in the context of sota models (that the baseline are well enough to make them interpretable).

**Strengths And Weaknesses:**

**Strengths**
- reproduction of the previously published paper (this is really good for community) with specifying and discussing issues, releasing reproduction code
- applying the method to the audio domain -- this is important step to know generalization of the technique to speed up adoption of the method if it is applicable widespread
- important discussion of the discrepancies between audio data and vision data, as well as issue of multi-modal representations

**Weaknesses**
- I believe more stressing and a bit better formatting of the text is needed, as well as some clarity is absent on some steps (see requested changes)
- issues on reproducing the original model -- I understand that authors of the prior paper didn't respond and maybe the training is not reproducible at all, but some numbers in Table 1 is super bad (30% accuracy on CIFAR) which causes doubts how meaningful then the reproduction of the PCBM then?
- Imagenet is used as pretraining data for audio data -- I am working a lot in speech domain and this is unexpected for me to see this kind of pretraining for audio classification to be honest. Why not some other data?
- in terms of reproduction -- it is not matched in the range of std, so I would be careful here in formulation as std within the same code base is small, while the std between the codebases is actually high, though the ordering between the methods preserved. Maybe some discussion on the std between the codebases is needed.

---

> ### Author Response · Authors · 2024-05-01
> **Comment reply**
>
> Dear Reviewer KhdV,
>
> We truly appreciate your review, especially your detailed and thorough advice. Here are our response to each of your comments and requested changes.
>
> ### Weaknesses
> > I believe more stressing and a bit better formatting of the text is needed, as well as some clarity is absent on some steps (see requested changes)
>
> **Response:** We have made edits as suggested.
>
> > Issues on reproducing the original model -- I understand that authors of the prior paper didn't respond and maybe the training is not reproducible at all, but some numbers in Table 1 is super bad (30% accuracy on CIFAR) which causes doubts how meaningful then the reproduction of the PCBM then?
>
> **Response:**
> We have modified our methodology: The original approach of CLIP original model leveraged CLIP for zero-shot classification, utilizing its pre-trained embeddings to directly match images with textual class descriptions without further training. Our adjusted approach, in contrast, involves training custom classifiers on top of CLIP’s image features for specific tasks like binary and multi-class classification. This enhances clip baseline model performance:
>
> | DATASET   | ACC   |
> |-----------|-------|
> | Cifar10   | 0.879 |
> | Cifar100  | 0.691 |
> | Coco      | 0.793 |
>
> Now they align with the original paper’s results, and these results have been updated in the paper accordingly. Thank you for pointing this out.
>
> > Imagenet is used as pretraining data for audio data -- I am working a lot in speech domain and this is unexpected for me to see this kind of pretraining for audio classification to be honest. Why not some other data?
>
> **Response:**
> Though it is true that this sounds quite strange and would not be recommended for an audio-task where performance was cruicial, this was done to create a baseline to compare audio models with, performing relatively well after fine-tuning. We do not have access to enough resources to train a model from scratch, though we are currently conducting tests with wav2vec as suggested (and we will attempt to add this into final manuscript). However, we believe this still serves our purpose to verify the stated claims of the original paper.
>
> > in terms of reproduction -- it is not matched in the range of std, so I would be careful here in formulation as std within the same code base is small, while the std between the codebases is actually high, though the ordering between the methods preserved. Maybe some discussion on the std between the codebases is needed
>
> **Response:**
> The std mismatches were due to calculation errors on our part. They are now fixed.
>
> ### Requested Changes: Clarity
> > page 2 Claim 1 formulation: "any neural network" - this is too strong claim, as we cannot brute force all possible models, I would smooth formulation. Otherwise did original paper had math proof (which I guess also impossible to have with the current math we have)
>
> **Response:**
> The original paper states this claim verbatim. We agree with your opinion and have modified it to a softer claim for the purposes of our study in the final manuscript. Indeed we cannot *prove* the claim but we aim to strenghten it by evaluating whether we can turn neural networks in other (untested) domains into PCBMs, e.g. audio classification.
>
> > page 2 claim 2 formulation: "original model" - it is not clear for the entire paper what is exactly original model - is it model we train which later we try to change to the PCBM by changes the arch and loss and training from scratch? Maybe it is CBM model?
>
> **Response:**
> The *original model* indeed refers to the models that are turned into PCBMs (as mentioned in claim 1). This has been reformulated to be clearer in the final manuscript.
>
> > sec 2.1 "in both domains" - you are speaking about audio, why here both domains? fix paragraphs on speaking only about vision, then about audio, then about both together.
>
> **Response:**
> Thank for for the note; we have rearranged the paragraphs so it reads clearer.
>
> > through the paper for me it is not clear which parts are reproduction, which parts some modifications. Better to be strict everywhere on this and have bold text e.g. About audio part - be also clear in text that all things are new and not reproduction. Right now it is mixed a bit in the text.
>
> **Response:**
> We have now bolded text in the abstract and in introduction that to make it clearer that all experiments w.r.t. audio is extension, and made a note at the top of the the methods section.
>
> > sec 2.3 f(x) definition is not given
>
> **Response:**
> It is defined as the embeddings of the data (extract from the text: *...are the projections of data embeddings $f(x)$ onto the concept subspace $C$...*). We have clarified it further in the final manuscript.
>
> > what is exactly "original model"? from text it is hard to get sec 3.2 last paragraph - highlight it more - it is important thing you checked!
>
> **Response:**
> We have changed the wording to ‘backbone model’ to be clearer.

---

> ### Author Response · Authors · 2024-05-01
> **cont'd**
>
> > sec 2.4 better to discuss why you decided to discard COCO right away in the section. Also is the size of COCO benchmark is larger? I wonder if there are issues on reproduction the scaled version, when data/model size is increased.
>
> **Response:**
> In the original paper they create subset to reduce the required disk space for experimentation. We have now adapted the same approach.
>
>
>
> > Table 3: first sentence in the caption is really weird, I don't see the "half the accuracy loss"
>
> **Response:**
> This is now Table 4. We intended it to mean that the PCBM-h improves performance by half the deficit that the PCBM had with respect to the original model. We agree this wording is clunky. We have simplified it to: *the PCBM trained in the audio domain performs almost as well as the baseline ResNet34 while the PCBM-h decreases
> the gap in accuracy further*.
>
> > page 8 top paragraph - "any neural net can be" - this is too strong claim, we never can show it (except from theoretical proof). So you need to smooth formulation.
>
> **Response:**
> Agreed and discussed earlier in this rebuttal. Reformulated into ''a wide range of neural networks''
>
> > Sec 4.3 last paragraph "for sounds We believe" - missed dot (I didn't see any other typos :)
>
> **Response:**
> Fixed. Thanks for pointing this out!
>
> > Sec 4.5 - "spectrogram image data" - remove image, why it is there?
>
> **Response:**
> This has been corrected.
>
> > Appendix B "It is repeated 50 times" - I didn't get what is exactly repeated. Why also in the prompt it is underscore in the names of classes? Did you try to remove it to get better output as we know things are sensitive to the punctuation formatting in the prompts.
>
> **Response:**
> We agree that this figure induces more confusion than clarity. We have removed this and instead described our process in more detail in section 2.2.
>
> ### Requested Changes: Methodological
> > why did you use Imagenet as pretrained model to finetune on audio dataset? I think this is really inappropriate comparison
>
> **Response:**
> We used Imagenet as a pretrained model to make use of transfer learning to speed up the training process of the network. Similar to what was done here (https://wandb.ai/jhartquist/fastaudio-esc-50/reports/Fine-Tuning-ResNet-18-for-Audio-Classification--VmlldzoyOTAyMzc), a pretrained resnet model was finetuned with the esc-50 dataset.
>
> > why for the second dataset of audio (ESC 50) you use pretrained model to be the one finetuned on FSDKaggle2018?
>
> **Response:**
> We used the pretrained model that was finetuned on the FSDKaggle2018 for the second dataset (esc 50) to add more audio data to training of the model. The FSDKaggle2018 dataset is a more general audio dataset which we used to finetune the resnet model first, then with this trained model we finetuned it again for the ESC 50 dataset which is specifically for environmental sounds.
>
> > the std for the same codebase is small, but reproduction between codebases is not within this std, so discussion on why there are still discrepancies would be beneficial
>
> **Response:**
> We have expanded upon this in section 4.2 in the new version of the paper.
>
> > figure 3 - is it because we used pretrained model on Imagenet?
>
> **Response:**
> In theory, we think it might be because of architecture but it can also be the dataset the model is trained on.
>
> > why not using AudioSet for experiments? It is more popular dataset, maybe the issue it is big for resources you have to be able to train on it? (then it is ok if not enough resources).
>
> **Response:**
> Due to lack of resources, in particular memory issues, we were unable to train the models on AudioSet
>
> > why any model trained on audio / speech is not used as pretrained model? e.g. trying wav2vec 2.0 or some other things from speech domain - yep it is OOD for the audio classification but closer in terms of representation that imagenet models.
>
> **Response:**
> Because of time constrains, we were unable to test wav2vec or another model from the speech domain. However with the HTSAT transformer model we used the pretrained model trained on audio which we finetuned for the esc-50 dataset.
>
> > some prior works on audio datasets should be provided to get sense how the original model performance or proposed model perform in the context of sota models (that the baseline are well enough to make them interpretable).
>
> **Response:**
> For the audio datasets no prior work about different model performances were provided because the comparison of performance was not necesserilay between two models on the same audio dataset, but rather with the model original performance and the same model performance with pcbm of pcbm-h. The accuracy of the original model can then be seen as a baseline where the pcbm and pcbm-h versions of the model are compared to.
>
> Thank you for your time and your review!

---

> > ### Comment · Reviewer_KhdV · 2024-05-07
> > **Reply to authors comments**
> >
> > Dear authors,
> >
> > Thanks for detailed clear responses and fixing the issues in the text and experimentation! Could you confirm that the revised version is uploaded and I can go over it once more time to check modifications you made?
> >
> > With respect to the SOTA models and pre-training on audio/speech dataset instead of imagenet - I got your point on the limited computations. It is a valid point and I am on your side on that. However, I still believe and think comparison should be done with the models pretrained on AudioSet or speech, e.g. wav2vec or any other pre-trained models published in speech / audio domain (I am ok with any reasonable model having your compute constrains, so that you only fine-tune it).
> >
> > I got that you prepared the paper for reproducibility challenge, however now in the discussion period we have time to adjust a bit the experimentation to make it proper for audio part. Also reproducibility is not about new additional benchmark you propose if you focus only on the reproducibility challenge I believe -- please correct me here if I am wrong with respect to policy on the reproducibility challenge. So either you focus only on reproducibility or make proper audio baselines (at least this is my personal subjective view) for the final revision.
> >
> > Reviewer.

---

> ### Author Response · Authors · 2024-05-11
> **Reply to reviewer's comment**
>
> Dear Reviewer,
>
> Thank you very much for your suggestions. The revised version has already been uploaded.
>
> Essentially the goal of the audio extension is to verify the claims of the authors. As in earlier works (e.g. https://openreview.net/pdf?id=eJmQJT0Dtt) the goal was just not to reproduce the paper, but to also verify the broader claims and generalizability of the method to make sure what is being published is useful. Therefore the submission is not meant to introduce PCBMs to the audio domain alone, rather it is meant to show that PCBMs may be useful to researchers in audio DL or other domains who may wish to use it. Thus, it is just a pilot experiment to show that it is feasible.
>
> For future work, it would be a reasonable next step to try Audioset and wav2vec. But for now, we believe the work that has been done so far should be sufficient for the broader objective of the paper.
>
> Best Regards,
>
> Authors

---

### Review · Reviewer_zgBC · 2024-03-30

**Summary Of Contributions:**

This paper mainly investigates the reproducibility of post-hoc concept bottleneck models (PCBM) and explores their application to the audio classification task. The paper first describes the PCBMs and their variants (Hybrid PCBMs (PCBM-h)) and shows their reproduction attempts. Their experiments find differences from the original paper and discuss the potential difference in their reproduction. The application experiments on the audio domain and found its concept list differs depending on the choice of the base model.

**Audience:**

No

**Broader Impact Concerns:**

No concern.

**Claims And Evidence:**

No

**Requested Changes:**

- The authors should focus more on making sufficient technical novelties instead of only focusing on reproduction or application to the other domains, which will attract more TMLR readers.
- If the authors stick to reproduction, they should at least make more efforts to use the same experimental configurations (e.g., by contacting the authors of the PCBM paper).

Other minor comments:
- The abstract should be more concrete. The abstract requires specific knowledge about Concept Bottleneck Models (CBMs) and Post-hoc Concept Bottleneck Models (PCBMs). It is difficult for general ML readers to understand this abstract.
- Figure 4 seems to be vital information to discuss in Section 3.4. So, moving Figure 4 to the main body is better than the appendix section.

**Strengths And Weaknesses:**

Strengths
- Reproduction of the other researchers' reports is essential in general.
- Application to the other domain (audio domain) shows the robustness of the PCBM techniques.

Weaknesses
- Very weak technical novelty. The paper mainly focuses on reproduction, and no significant technical novelty exists.
- Although reproduction is essential, the current experiments have several unclear parts compared with the original authors' experimental configurations, and it is difficult to claim the findings about reproduction and application to the audio domain from this experimental result.

---

> ### Author Response · Authors · 2024-05-01
> **comment reply**
>
> Dear Reviewer zgBC,
>
> Thank you very much for your feedback and acknowledgement of our paper's strengths! We appreciate the valuable feedback provided on our paper. Below are our response to your comments. The minor errors have also been corrected them as suggested.
>
> ### Weakness
> > Very weak technical novelty. The paper mainly focuses on reproduction, and no significant technical novelty exists.
>
> **Response:**
> The objective of our paper is to test the reproducibility of the original study on PCBMs, including the reproducibility of its original results and its generalizability. Our work has been submitted to ML reproducibility challenge, which is in collaboration with TMLR.  We believe our work is in line with the spirit of a typical reproducibility paper.
>
> > Although reproduction is essential, the current experiments have several unclear parts compared with the original authors' experimental configurations, and it is difficult to claim the findings about reproduction and application to the audio domain from this experimental result.
>
> **Response:**
> For the reproduction of the original results, we have made a few improvements and corrections, which made our results more comparable to those from the original study. The differences between the original and our results are now within 5%. Firstly, the experiments on COCO-stuff have been added. The results can be found in table 1 and 2. They align with the original paper’s results; Secondly, the performance of original model CLIP has been improved (the result can be found in tables 1 and 2) and is now close to the original results. Finally, we corrected the std’s calculation error in Table 1 and Table 2.
>
> For the extension to the audio domain, due to the time constraints on the resubmission deadline and computational resources, we have not finished all the work on AudioSet. However, we believe our current work has laid a foundation for future work and provides preliminary evidence on the generalisability of PCBMs .
> <!-- <span style="color:red">to resolve word2vec</span> -->
>
> ### Requested Changes
> > The authors should focus more on making sufficient technical novelties instead of only focusing on reproduction or application to the other domains, which will attract more TMLR readers.
>
> **Response:**
> Since this is a  paper under the category of reproduction, we need to make sufficient effort in the reproduction. We believe reproduction is also very relevant to TMLR readers.
>
>
> > If the authors stick to reproduction, they should at least make more efforts to use the same experimental configurations (e.g., by contacting the authors of the PCBM paper).
>
> **Response:**
> We did contact the authors but did not get replies. Therefore we made educated guess on some of the experimental configurations (e.g. default hyperparameters for some packages). We declare all the hyperparameters in our paper and the code. For the reproduction of the original results, we have made a few improvements and corrections, which made our results more comparable to those from the original study. The differences between the original and our results are now within 5%.
>
> Thank you for your time and your review!

---

> > ### Comment · Reviewer_zgBC · 2024-05-17
> > **Thanks for the clarification**
> >
> > Thanks for the clarification, and I apologize for misunderstanding your submission category.
> > Then, the paper matches the TMLR's scope.
> > Also, it is unfortunate that the authors did not respond to you, and I appreciate your efforts in making the experimental configurations close to this reference paper.

---

### Review · Reviewer_iwDj · 2024-04-17

**Summary Of Contributions:**

This submission a) reports on the reproduction of experiments conducted by Yuksekgonul et al. in the paper titled "Post-hoc Concept Bottleneck Models" (PCBM), b) extends the underlying approach to audio classification tasks, and b) conducts a similar experimental investigation in two audio classification tasks. One of the purposes for conducting steps b) and c) is to verify the claim made by the original paper authors that any NN can be converted in high-performing PCBM.

**Audience:**

Yes

**Claims And Evidence:**

No

**Requested Changes:**

Either produce a higher quality reproduction investigation (I believe there are examples of published reproduction papers in ML and more generally (e.g. https://rescience.github.io/)), or reduce the amount of content devoted to that and increase the part linked with the audio classification task (and maybe some other ones.)

Provide a more comprehensive investigation into applying PCBMs in audio classification tasks.

**Strengths And Weaknesses:**

Strengths
CBMs and their variants provide one of many approaches for interpreting and understanding black-box AI systems. Attempts at reproducing prior results and extending this work to new domains can be commended.

Weaknesses
Unfortunately I have serious reservations regarding both the reproduction of prior results as well as the extension of PCBMs to new domains.

Your manuscript fails to mention what information you had available in order to start reproducing the results published in the original paper. You provide no discussion about the likelihood of being able to reproduce results nor you discuss possible impacts of not having any of the ingredients required. You do not define what you mean by successful reproduction. It seems in your manuscript you are equally happy with differences as small as 0.008 and as large as 0.037. The huge discrepancy in reproducing original models is very concerning yet apart from some hypothetical guesses you offer no other explanation. Given that some of the key ingredients required to reproduce the results were missing from the start, I find your reproduction experiments offering limited value. It would have sufficed to simply state that this is the best you could get given information available to you and focus on adding content to the manuscript from elsewhere (such as new domains, new (P)CBM variants, etc.).

Your investigation of PCBMs for audio classification is quite limited in its scope and options you have examined. You have not provided any explanation of what other options other than ChatGPT were there to derive meaningful categories. Neither you described how meaningful it is to use (clearly) suboptimal categories and what kind of impact this choice has on the PCBMs and their use.

---

> ### Author Response · Authors · 2024-05-01
> **comment reply**
>
> Dear Reviewer iwDj,
>
> Thank you very much for your feedback and acknowledgement of our paper's strengths! We appreciate the valuable feedback provided on our paper. Below are our response to your comments.
>
> ### Weakness
> > Your manuscript fails to mention what information you had available in order to start reproducing the results published in the original paper. You provide no discussion about the likelihood of being able to reproduce results nor you discuss possible impacts of not having any of the ingredients required. You do not define what you mean by successful reproduction. It seems in your manuscript you are equally happy with differences as small as 0.008 and as large as 0.037. The huge discrepancy in reproducing original models is very concerning yet apart from some hypothetical guesses you offer no other explanation. Given that some of the key ingredients required to reproduce the results were missing from the start, I find your reproduction experiments offering limited value. It would have sufficed to simply state that this is the best you could get given information available to you and focus on adding content to the manuscript from elsewhere (such as new domains, new (P)CBM variants, etc.).
>
> **Response:**
> For the past papers winning ML reproducibility challenge, we did not find a standard way to quantify the success of reproduction. In our paper, we aim at confirming the trend on which PCBM-h performs better than PCBMs. We also measured the differences between the results from original work and our reproduction, which are within 5% (further analysis would be required in the future. also see reference in the paper on why 5% is an acceptable threshold)
>
> > Your investigation of PCBMs for audio classification is quite limited in its scope and options you have examined. You have not provided any explanation of what other options other than ChatGPT were there to derive meaningful categories. Neither you described how meaningful it is to use (clearly) suboptimal categories and what kind of impact this choice has on the PCBMs and their use.
>
> **Response:**
> For the extension to the audio domain, we aimed to provide a preliminary benchmark and future directions to work with which we believe we archived. Due to the time constraints on the resubmission deadline and computational resources, we have not finished all the work on AudioSet. However, we believe our current work has laid a foundation for future work and provides preliminary evidence on the generalisability of PCBMs .
>
> ### Requested Changes
> > Either produce a higher quality reproduction investigation (I believe there are examples of published reproduction papers in ML and more generally (e.g. https://rescience.github.io/)), or reduce the amount of content devoted to that and increase the part linked with the audio classification task (and maybe some other ones.)
>
> **Response:**
> For the reproduction of the original results, we have made a few improvements and corrections, which made our results more comparable to those from the original study. The differences between the original and our results are now within 5%. Firstly, the experiments on COCO-stuff have been added. The results can be found in table 1 and 2. They align with the original paper’s results; Secondly, the performance of original model CLIP has been improved (the result can be found in tables 1 and 2) and is now close to the original results. Finally, we corrected the std’s calculation error in Table 1 and Table 2.
>
> For the extension to the audio domain, due to the time constraints on the resubmission deadline and computational resources, we have not finished all the work on AudioSet. However, we believe our current work has laid a foundation for future work and provides preliminary evidence on the generalisability of PCBMs.
>
> > Provide a more comprehensive investigation into applying PCBMs in audio classification tasks.
>
> **Response:**
> Due to the time constraints on the resubmission deadline and computational resources, we cannot provide further results. However, the current work should serve as a benchmark and enlighten future research, which is the main goal of our work.
>
> Since this is a reproduction paper, the aim of the extension to audio domain is to test the claims of the original paper including that PCBMs are applicable to different neural networks.
>
>
> Thank you for your time and your review!

---

### Review · Reviewer_bDZM · 2024-04-17

**Summary Of Contributions:**

This paper investigates the prior work on Post-hoc Concept Bottleneck Models (PCBMs) and their ability to enhance the interpretability of deep neural networks while maintaining high performance. This involves applying the PCBM approach to image classification tasks using the same datasets and backbone models as the original study. The authors reproduce key experiments from the original PCBM paper in the image domain and extend the approach to audio classification while evaluating three main claims: 1: Applicability to any neural network. 2: Comparable performance to the original model. 3: Interpretability benefits via this approach. Moreover, for the audio, the process first involves adapting the approach to work and then generate audio-specific concept banks, which enables the evaluation of performance on two audio classification datasets.

Authors find that PCBMs maintain high accuracy with only a small performance drop compared to the baseline models, supporting claims 1 and 2. However, the interpretability of the ResNet34 PCBM is questionable, as the identified impactful concepts do not intuitively match the class labels. In contrast, the HTS-AT PCBM demonstrates better alignment between concepts and classes, suggesting the importance of embedding alignment.

**Audience:**

Yes

**Broader Impact Concerns:**

No concern.

**Claims And Evidence:**

No

**Requested Changes:**

The paper should strive for a higher quality reproduction study with a detailed methodology and analysis of discrepancies or reduce the emphasis on reproduction and dedicate more attention to novel contributions. Experimental analysis for the audio classification task should be conducted using AudioSet for comparison of results with other approaches.

Embedding misalignment issues should be carefully studied with the differences in architectures to be ruled out as that can cause discrepancies in the embedding spaces leading to incorrect concept

**Strengths And Weaknesses:**

**Strengths**:

* Thoroughly investigates PCBMs and their claims, providing valuable insights into their performance and interpretability.
* Extends the approach to the audio domain, demonstrating its potential for broader application.
* Discusses challenges and limitations encountered during reproduction and analysis, promoting transparency and reproducibility in research.
* Identifies that there can be differences and incoherency in the concepts that are learnt vs human understanding of the audio signals.

**Weakness**:
Embedding misalignment issues: The paper highlights the potential misalignment between embeddings generated by the backbone model (e.g., ResNet34) and the audio encoder within the CLAP model used for concept generation. This difference in architectures may lead to discrepancies in the embedding spaces, resulting in inaccurate or irrelevant concepts being associated with the audio inputs.

---

> ### Author Response · Authors · 2024-05-01
> **comment reply**
>
> Dear Reviewer bDZM,
>
> Thank you very much for your feedback and acknowledgement of our paper's strengths! We appreciate the valuable feedback provided on our paper. Below are our response to your comments.
>
> ### Weakness
> > Embedding misalignment issues: The paper highlights the potential misalignment between embeddings generated by the backbone model (e.g., ResNet34) and the audio encoder within the CLAP model used for concept generation. This difference in architectures may lead to discrepancies in the embedding spaces, resulting in inaccurate or irrelevant concepts being associated with the audio inputs.
>
> **Response:**
> Our work aims at testing the reproducibility of the original study on PCBMs. To be objective, we take a neutral ground with both the advantages and disadvantages of PCBMs. By identifying the weaknesses of the original work, we would like to evoke discussions and warn researchers about the potential pitfalls of the use of PCBMs in the future. We believe that identifying those weaknesses themselves should not be considered as the weakness of our work, but rather a contribution of our work.
>
> ### Requested Changes
>
> > The paper should strive for a higher quality reproduction study with a detailed methodology and analysis of discrepancies or reduce the emphasis on reproduction and dedicate more attention to novel contributions. Experimental analysis for the audio classification task should be conducted using AudioSet for comparison of results with other approaches.
>
> **Response:**
> The objective of our paper is to test the reproducibility of the original study on PCBMs, including the reproducibility of its original results and the generalizability of the claims and methods to other domains. Our work is in line with the spirit of a typical reproducibility paper, which falls squarely in line with TMLR's objectives (https://jmlr.org/tmlr/editorial-policies.html). The Machine Learning Reproducibility Challenge (https://reproml.org/) is in collaboration with TMLR this year for precisely this purpose.
>
> For the reproduction of the original results, we have made a few improvements and corrections, which made our results more comparable to those from the original study. The differences between the original and our results are now within 5% points for each result. As for additions to the manuscript that may help based on your and other reviews, firstly, the experiments on COCO-stuff have been added. The results can be found in table 1 and 2. They align with the original paper’s results; Secondly, the performance of original model CLIP has been improved (the result can be found in tables 1 and 2) and is now close to the original results. Finally, we corrected the std’s calculation error in Table 1 and Table 2. All in all, we show that the paper is reproducible to a small error margin.
>
> For the extension to the audio domain, due to the time and resource constraints, we have not finished all the work on AudioSet. However, we believe our current work has laid a foundation for future work and provides preliminary evidence on the generalisability of PCBMs. We hope this response suffices to aleviate your concerns.
>
> > Embedding misalignment issues should be carefully studied with the differences in architectures to be ruled out as that can cause discrepancies in the embedding spaces leading to incorrect concept
>
> **Response:**
> Our paper is not try to defend PCBMs but rather to evaluate them on a neural ground. The embedding misalignment issues are what we hypothesied and we would like to bring this to the attention of the readers in our discussion and provide a direction for future research.
>
> Thank you for your time and your review!

---

> > ### Comment · Reviewer_bDZM · 2024-05-17
> > **Thank you for the clarification**
> >
> > Thank you for providing the rationale and clarifying the position for the paper. Thank you for your detailed and thoughtful responses to my feedback. I appreciate the effort you have put into addressing the points raised, and I acknowledge the strengths and contributions of your work.
> >
> > Reproduction Study Quality: Your clarification that the paper's objective is to test the reproducibility of the original study on PCBMs is well-noted. The improvements and corrections made to achieve a closer alignment with the original results demonstrate your commitment to a rigorous reproduction effort. The additional experiments on COCO-stuff and the improved performance of the original model CLIP, along with the corrected std’s calculation error, contribute significantly to the robustness of your study.
> >
> > I appreciate your transparency regarding the time and resource constraints that limited the completion of work on AudioSet. Your current work lays a solid foundation for future research and provides preliminary evidence on the generalizability of PCBMs, which is valuable.
> >
> > Your hypothesis on embedding misalignment issues and your intention to bring this to the readers' attention for future research directions are commendable. Exploring these discrepancies is crucial, and your work rightly opens the door for further investigation in this area.
> >
> > Overall, your responses satisfactorily address my concerns and contribute to the ongoing discourse on PCBMs.

---

### Decision · Action_Editor_eGSz · 2024-06-12

**Recommendation:** Reject

**Comment:**

Overall, I recommend rejection, with the possibility of a major revision, for the reasons I outlined in the "Claims And Evidence" section.

**Audience:**

Yes, I believe at least some individuals in TMLR's audience would be interested in this paper's findings.

**Claims And Evidence:**

The primary goal of this submission is to reproduce the ICLR'22 paper "Post-hoc concept bottleneck models" by Yuksekgonul et al. Moreover, the authors apply post-hoc concept bottleneck models (PCBMs) to the audio domain (Yuksekgonul et al. applied it only to vision data).

Overall, the authors found that the paper was largely reproducible. The main difference was that in the initial submission, the baseline models did not achieve the same performance that Yuksekgonul et al. claimed (e.g., Yuksekgonul et al.'s baseline model obtained 0.7 accuracy on CIFAR100, whereas the submission obtained 0.31 accuracy, which is a big difference). However, the accuracy results for PCBMs were more comparable. In the revision, the authors got the baseline accuracy much higher.

However, I am recommending a rejection, with the possibility of a major revision, since there are major concerns among the reviewers about the authors' methodologies, claiming that the authors do not "follow good practices established in the paper reproduction community" and that "the authors' own practices [may not be] methodical enough to convince others, like me, that their reproduction approach is flawless."

Moreover, there are concerns with the application to the audio domain. Reviewers with domain expertise were surprised the authors used ImageNet for pertaining rather than wav2vec or any other pre-trained models published in the speech / audio domain. The authors responded that they did not due to time constraints. Since there is no deadline for the major revision and resubmission, I recommend the authors take the time to make the experiments more compelling for the audio domain.

**Resubmission Of Major Revision:**

The authors may consider submitting a major revision at a later time.